# An ADMM Based Parallel Approach for Fund of Fund Construction

Yidong Chen [1,2,†], Chen Li [1,†] and Zhonghua Lu [1,*]

1 Computer Network Information Center, Chinese Academy of Sciences, Beijing 100190, China; chenyidong@cnic.cn (Y.C.); lichen@sccas.cn (C.L.)
2 University of Chinese Academy of Sciences, Beijing 100049, China
* Correspondence: zhlu@cnic.cn
† These authors contributed equally to this work.

**Abstract:** In this paper, we propose a parallel algorithm for a fund of fund (FOF) optimization model. Based on the structure of objective function, we create an augmented Lagrangian function and separate the quadratic term from the nonlinear term by the alternate direction multiplier method (ADMM), which creates two new subproblems that are much easier to be computed. To accelerate the convergence speed of the proposed algorithm, we use an adaptive step size method to adjust the step parameter according to the residual of the dual problem at every iterate. We show the parallelization of the proposed algorithm and implement it on CUDA with block storage for the structured matrix, which is shown to be up to two orders of magnitude faster than the CPU implementation on large-scale problems.

**Keywords:** FOF construction; non-linear optimization; ADMM algorithm; parallel computing; GPU

## 1. Introduction

Fund of funds (FOF) has become a hot topic during the past several years. As a mutual fund scheme, FOF uses other funds as investment targets and achieve the purpose of indirectly holding securities such as stocks and bonds [1]. The first target of FOF construction is to obtain optimal portfolio among different funds based on the tradeoff between return and risk. To meet this goal, one of the main research activities for the past few years has been FOF modeling. It is worth mentioning that many existing FOF optimization models are based on the Mean-Variance (MV) framework. In 1952, Markowitz introduced the seminal work, the MV model [2], which is regarded as the fundamental finding of modern portfolio theory (MPT). Since then, it has become the most essential theory to study the FOF asset allocation problem. Despite huge attention, the MV model still has some obvious shortcomings. One drawback which many studies emphasize is that the optimal portfolio can be extremely sensitive to input parameters [3,4]. Another drawback of the MV model is that this approach sometimes provides over-concentrated portfolios, which would probably cause huge losses if a financial crisis were to happen. Moreover, the MV model ignores the subjective views of investors. These shortcomings limit the MV models' application in the FOF construction. As a result, it is not surprising that many research efforts were placed on extending the MV model [5–11].

From a practical perspective, there exist many trading restrictions in the real-world financial market. Transaction cost is one of which that should be paid attention. In fact, transaction costs in the financial market is usually treated as non-linear functions, which makes the FOF optimization model a nonlinear programming problem. The interior point method (IPM) [12,13], sequence quadratic programming (SQP) method [14], parallel variable distribution (PVD) [15] and line search algorithm [16] are all popular methods for nonlinear programming problems. However, the aforementioned methods can be inefficient since they does not explore the structure of the objective function. For example, the general

SQP procedure uses the derivative vector and the Hessian matrix of the Lagrangian function to construct quadratic approximation at each iteration [17]. The Hessian matrix of FOF optimization models may not necessarily be positive semi-definite, so an approximation of the Hessian matrix is needed. Constructing a Hessian approximation by quasi-Newton methods may be poorly performed and time-consuming. The IPM maintains the feasibility during iterations but it is a centralized and computationally expensive method. Moreover, it is hard to parallelize. The PVD method is suitable for parallelization but needs to solve a difficult convex subproblem when applied to FOF optimization model, which makes it hard to be extended to large-scale problems.

Most of algorithms that developed in recent years have concentrated on structured nonlinear programming, which aims to characterize a range of nonlinear problems. In 1977, Glowinski and Marrocco [18] proposed the alternating direction method of multiplies (ADMM) based on the decomposition. It is widely used in large-scale nonlinear optimization problems thanks to the advantage of being easily extended to parallel and distributed systems. However, it still faces the challenge of slow convergence when applied in the FOF optimization model. Chaves et al. [19] first proposed a Newton-based efficient algorithm; however, it only works when facing a non-constrained FOF model, which is unrealistic in practice. A least-squares method and an alternating direction method were developed by Bai et al. [20] and Costa et al. [21], respectively. However, they still consume a lot of time as the scale of the problem grows.

The main contribution of this paper is to propose a parallel algorithm general enough to characterize most of the existing specific FOF optimization models. Moreover, our approach is able to solve a large number of FOF optimization models that have the same structure as this paper studied. The main contributions of this paper could be summarized as follows:

- We develop a new FOF optimization model which integrates complicated and diversified constraints into the Mean-Variance framework accompanied by a Black–Litterman based-asset expected return and covariance.
- We propose an approach to handle the FOF optimization model mentioned above based on elevating the original problem into a higher-dimension convex problem which is solved by a modified ADMM algorithm. Moreover, we compare the modified ADMM algorithm efficiency to two of the best performing methods, SQP and IPM.
- To explore a larger search space for better solutions, we parallelize the proposed approach on GPU using CUDA and study its speedup on different problem scales.

The remainder of the paper is organized as follows. In Section 2, we give the step-by-step formulation of our new FOF optimization model, especially focusing on the Black–Litterman based-asset expected return and covariance. Section 3 contains an approach to handle the FOF optimization model mentioned above. More specifically, we introduce new variables to transfer the inequality constraints in the model to the equality constraints and use a modified ADMM algorithm to construct the optimal portfolio. In Section 4, the GPU-based parallel approach using CUDA is introduced to improve computational efficiency and cope with large scale problems. This is followed by implementation details of the proposed parallel approach in Section 5,where we present the efficiency of the proposed approach compared with some of the best performing methods, as well as the acceleration effect of the parallel approach. In Section 6, we conclude the paper.

## 2. Problem Formulation

The core of FOF funds is to diversify investment, so it is no surprise that asset allocation has become an important part of FOF construction. Based on the asset allocation, we can improve the return-risk feature of a fund portfolio. For simplicity, we limit the fund type to equity funds, so asset allocation is beyond the scope of this paper. Furthermore, another determining factor for the performance of an FOF strategy is the quality of the fund pool. It is naturally an extremely important thing for fund managers to screen out their own fund pools from a huge number of funds and build fund portfolios accordingly. There are

many fund selection criteria such as risk/return parameters and the professionalism of the management team. In this paper, we take return, standard deviation and fund size as the key factors of focus to screen out funds.

It is assumed that investors with initial capital $C$ will assign their wealth to $n$ equity funds in the following $T$ periods. Let $x_i^0$ be the proportion on fund $i$ included in the portfolio at the previous period. $S_r$, $S_s$, $S_m$, $S_b$ and $S_c$ are the set of the funds with high risk and high return, with medium to high risk and medium to high return, with medium risk and medium return, with low to medium risk and low to medium return, and with low risk and low return, included in the portfolio, respectively. A new FOF optimization model is proposed as follows:

$$
\begin{aligned}
\min_{x \in \mathbf{R}^n} \quad & \tfrac{1}{2} x^T P x + q^T x + \sum_{i=1}^n f_i(x_i) \\
\text{subject to} \quad & \sum_{i \in S_r} x_i \le 0.2 \\
& \sum_{i \in S_c} x_i \le 0.15 \\
& 0.95 \sum_{i \in S_s} x_i + 0.6 \sum_{j \in S_m} x_j \le 0.4 \\
& \sum_{i \in S_b} x_i + 0.3 \sum_{j \in S_m} x_j \ge 0.4 \\
& \mathbf{1}^T x = 1 \\
& l_i \le x_i \le u_i, i = 1, \dots, n
\end{aligned}
\tag{1}
$$

where the transaction cost $f_i(x) = \exp(-(\frac{C \max\{x_i - x_i^0, 0\} + a_i}{b_i})^2)$, $a_i \in \mathbf{R}$ and $b_i \in \mathbf{R}$ are the given parameters, which are various from different funds. $P$ denotes the covariance matrix of the fund returns, and $q \in \mathbf{R}^n$ refers to the fund returns vector. $l_i$ and $u_i(i = 1, \dots, n)$ are the lower and upper bound of investment proportion on fund $i$. Without loss of generality, we can abstract the new FOF optimization model into a general form.

Notation. Let $\mathbf{R}$ denote the set of real numbers, $\mathbf{R}^n$ the $n$-dimensional real space, and $\mathbf{S}_{++}^n (\mathbf{S}_+^n)$ the set of real $n$-by-$n$ symmetric positive (semi)definite matrices. We denote by $\mathbf{R}^{m \times n}$ the set of real $m$-by-$n$ matrices, $I_n \in \mathbf{R}^{n \times n}$ is the identity matrix and $\mathbf{0} \in \mathbf{R}^{n \times n}$ is the zero matrix, and $\mathbf{1}_n$ is the $n$-dimensional vector with all the entries being 1. $\mathcal{I}_{\mathcal{C}}$ is the indicator function over the affine constraints of (5), i.e.,

$$
I_{\mathcal{C}}(x) = \begin{cases} 0 & x \in \mathcal{C} \\ \infty & \text{otherwise} \end{cases}
\tag{2}
$$

For $a, b \in \mathbf{R}$, the projection of $z$ onto $[a, b]$ is given by $\Pi[a, b](z) = \min(\max(z, a), b)$. The FOF optimization model that considered transaction cost is given by:

$$
\begin{aligned}
\min_{x \in \mathbf{R}^n} \quad & \tfrac{1}{2} x^T P x + q^T x + \sum_{i=1}^n f_i(x_i) \\
\text{subject to} \quad & Ax \le b, \\
& l_i \le x_i \le u_i, i = 1, \dots, n \\
& \mathbf{1}^T x = 1
\end{aligned}
\tag{3}
$$

where $x \in \mathbf{R}^n$ is the optimization variable. $P \in \mathbf{S}_+^n$ is a symmetric semidefinite covariance matrix and $q \in \mathbf{R}^n$. $A \in \mathbf{R}^{m \times n}$ and $b \in \mathbf{R}^{\in m}$. $l_i$ and $u_i(i = 1, \dots, n)$ are the lower and upper bound. $f_i(x), x \in \mathbf{R}, i = 1, \dots, n$ is the transaction cost function. The function $f_i(x)$ is the subscription and redemption cost for adjusting the $i$th fund. If the transaction cost $f_i(x)$ is given by linear function or quadratic function, i.e., $f_i(x) = \alpha_i x$, then the optimization problem (3) is quadratic programming. Throughout this paper, we consider the transaction cost function $f_i(x)$ to be convex, so that the objective function is nonlinear and convex. The linear constraints imply the bound of the portfolio weight and the investment requirement of the portfolio weight.

*Transforming the Inequality to Equality*

To solve the nonlinear problem (1), we consider the general formulation of the nonlinear problem (3). We start by transforming the convex problem (3) to a new problem without inequality constraints by introducing the slack variables to transfer the inequality constraints into the equality constraints. Let $\hat{x} \in \mathbf{R}^m$, the inequality constraints $Ax \leq b$ and the equality constraint $\mathbf{1}^T x = 1$, which could be rewritten as

$$\begin{bmatrix} A & \mathbf{1} \\ \mathbf{1}^T & \mathbf{0} \end{bmatrix} \begin{bmatrix} x \\ \hat{x} \end{bmatrix} = \begin{bmatrix} b \\ 1 \end{bmatrix}, l_i \leq x_i \leq u_i, i = 1, \ldots, n \hat{x}_i \geq 0, i = 1, \ldots, m. \tag{4}$$

Let $\tilde{x} = \begin{bmatrix} x \\ \hat{x} \end{bmatrix}$, then the problem (3) could be rewritten as

$$\begin{aligned} \min_{x \in \mathbf{R}^{m+n}} \quad & \tfrac{1}{2} \tilde{x}^T \tilde{P} \tilde{x} + \tilde{q}^T \tilde{x} + \sum_{i=1}^n f_i(\tilde{x}_i) \\ \text{subject to} \quad & \tilde{A} \tilde{x} = \tilde{b}, \\ & l_i \leq x_i \leq u_i, i = 1, \ldots, n \\ & \tilde{x}_i \geq 0, i = n+1, \ldots, n+m., \end{aligned} \tag{5}$$

where

$$\tilde{P} = \begin{bmatrix} P & \mathbf{0} \\ \mathbf{0} & \mathbf{0} \end{bmatrix}, \tilde{q} = \begin{bmatrix} q \\ \mathbf{0} \end{bmatrix}, \tilde{A} = \begin{bmatrix} A & \mathbf{1} \\ \mathbf{1}^T & \mathbf{0} \end{bmatrix}, \tilde{b} = \begin{bmatrix} b \\ 1 \end{bmatrix}$$

By introducing the slack variable $z \in R^{m+n}$, the problem (5) is equivalent to

$$\begin{aligned} \min_{x \in \mathbf{R}^{m+n} z \in \mathbf{R}^{m+n}} \quad & \tfrac{1}{2} \tilde{x}^T \tilde{P} \tilde{x} + \tilde{q}^T \tilde{x} + \sum_{i=1}^n f_i(\tilde{z}_i) + \mathcal{I}_{\mathcal{C}_1}(x) + \mathcal{I}_{\mathcal{C}_2}(z) \\ \text{subject to} \quad & \tilde{x}_i - z_i = 0, i = 1, \ldots, n+m. \end{aligned} \tag{6}$$

with variables $x \in \mathbf{R}^{m+n}$ and $z \in R^{m+n}$. $\mathcal{C}_1 = \{x | \tilde{A}\tilde{x} = \tilde{b}\}$, $\mathcal{C}_2 = \{z | l_i \leq z_i \leq u_i,$ $i = 1, \ldots, n, z_i \geq 0, i = n+1, \ldots, n+m\}$. One can easily figure that the original problem (3) is transformed to problem (6) by two steps. The first step is the slack variable $\hat{x}$, which changes the inequality $Ax = b$ to the equality $\tilde{A}\tilde{x} = \tilde{b}$. The second step, with the help of the slack variable $z$, transforms the problem (5) to problem (6) without any inequality constraints remaining.

## 3. Algorithm for FOF Optimization Model
### 3.1. ADMM Steps

It is costly to solve the optimization problem (5) or (6) by calling the default solvers such as the fmincon(MATLAB). One key to solve the problem (3) is to separate the quadratic term and the nonlinear term, since there are effective solvers for quadratic problems. The augmented Lagrangian of (6) is

$$L(x, z, u) = \frac{1}{2} \tilde{x}^T \tilde{P} \tilde{x} + \tilde{q}^T \tilde{x} + \sum_{i=1}^n f_i(\tilde{z}_i) + \mathcal{I}_{\mathcal{C}_1}(x) + \mathcal{I}_{\mathcal{C}_2}(z) + \frac{\tau}{2} ||x - z + \frac{u}{\tau}||^2. \tag{7}$$

Then, we develop the ADMM used in every iteration with regard to the variables $x^k$ and $z^k$ as follows.

$$x^{k+1} = \arg\min_{\tilde{x} \in \mathcal{C}_1} \frac{1}{2} \tilde{x}^T \tilde{P} \tilde{x} + \tilde{q}^T \tilde{x} + \frac{1}{2} ||x - z^k + u^k||^2 + \mathcal{I}_{\mathcal{C}_1}(x), \tag{8}$$

$$z^{k+1} = \arg\min_{\tilde{z} \in \mathcal{C}_2} \sum_{i=1}^n f_i(\tilde{z}_i) + \frac{\tau_k}{2} ||x^{k+1} - z + u^k||^2 + \mathcal{I}_{\mathcal{C}_2}(z), \tag{9}$$

$$u^{k+1} = u^k + (x^{k+1} - z^{k+1}). \tag{10}$$

The update of the variables $x^k$ is $z^k$ is obvious. We minimize the augmented Lagrangian function. In practice, we find that it requires many more iterations on large problems. The convergence of the ADMM algorithm is guaranteed under fairly general assumptions [22,23]. Since the objective function is convex and the global solution exists, we have $x^k \rightarrow x^*$, $z^k \rightarrow z^*$ and $u^k \rightarrow u^*$. For faster convergence, we suggest to perform the relaxed ADMM algorithm. The relaxed ADMM iterates $x^k$, $z^k$ and $u^k$, for $k = 0, 1, \ldots$, by

$$x^{k+1} = \arg\min_{\tilde{x} \in \mathcal{C}_1} \frac{1}{2}\tilde{x}^T \tilde{P} \tilde{x} + \tilde{q}^T \tilde{x} + \frac{\tau_k}{2}||x - z^k + \frac{u^k}{\tau_k}||^2 + \mathcal{I}_{\mathcal{C}_1}(x), \qquad (11)$$

$$x^{k+1} = \gamma_k x^{k+1} - (1 - \gamma_k)z^k \qquad (12)$$

$$z^{k+1} = \arg\min_{\tilde{z} \in \mathcal{C}_2} \sum_{i=1}^{n} f_i(\tilde{z}_i) + \frac{\tau_k}{2}||x^{k+1} - z + \frac{u^k}{\tau_k}||^2 + \mathcal{I}_{\mathcal{C}_2}(z), \qquad (13)$$

$$u^{k+1} = u^k + \tau_k(x^{k+1} - z^{k+1}). \qquad (14)$$

Here, $u^k \in \mathbf{R}^{m+n}$ denotes the dual variables on iteration $k$. $(\tau_k, \gamma_k)$ are sequences of penalty and relaxation parameters. If the problem (3) is solvable, then the sequence $(x^k, z^k, u^k)$ converges to its primal-dual solution. On the other hand, if the problem is infeasible, then the sequence $(x^k, z^k, u^k)$ does not converge, but the sequence $(x^k - x^{k-1}, z^k - z^{k-1}, u^k - u^{k-1})$ always converges and can be used to certify the infeasibility of the problem [24].

Since the nonlinear function $f_i(x)$ is separable, we develop the ADMM-based algorithm and parallelize it on CUDA. The key point of ADMM iteration presented in (6) contains three steps: the solving of the linear equation, the minimum of the nonlinear problem (27) and the update of the dual variables $\lambda$.

*3.2. Solving the Reduced KKT System*

We describe how $x^k$ is updated with (11). Minimizing the augmented Lagrangian (11) involves solving the following equality-constrained least-squares problem:

$$\arg\min_{\tilde{x} \in \mathcal{C}_1} \quad \frac{1}{2}\tilde{x}^T \tilde{P} \tilde{x} + \tilde{q}^T \tilde{x} + \frac{\tau_k}{2}||x - z^k + \frac{u^k}{\tau_k}||^2, \qquad (15)$$

$$\tilde{A}\tilde{x} = \tilde{b}. \qquad (16)$$

Let $v^k = z^k - \frac{u^k}{\tau_k}$, the minimizing $\tilde{x}$ can be found by solving the following linear equation:

$$\begin{bmatrix} \tilde{P} + \tau_k I & \tilde{A}^T \\ \tilde{A} & 0 \end{bmatrix} \begin{bmatrix} x^{k+1} \\ v \end{bmatrix} = \begin{bmatrix} \tau_k v^k - \tilde{q} \\ \tilde{b} \end{bmatrix}. \qquad (17)$$

$\tilde{P}$ is a $(m+n) \times (m+n)$ matrix and $\tilde{A}$ is a $m \times (m+n)$ matrix, we need to solve the $2m + n$ linear equation. As $\tau_k$ goes to change, we have to solve the linear system at each iteration, which will prevent us from solving the target problem effectively. Since $\tau_k > 0$, the matrix $\tilde{P} + \tau_k I$ is positive defined. There exist orthogonal matrix $\tilde{Q}$ such that

$$\tilde{P} = \tilde{Q}^T \tilde{D} \tilde{Q} \qquad (18)$$

We have

$$\begin{bmatrix} I & 0 \\ -\tilde{A}\tilde{Q}^T(\tilde{D} + \tau_k I)^{-1}\tilde{Q} & I \end{bmatrix} \begin{bmatrix} \tilde{P} + \tau_k I & \tilde{A}^T \\ \tilde{A} & 0 \end{bmatrix} = \begin{bmatrix} \tilde{P} + \tau_k I & \tilde{A}^T \\ 0 & -\tilde{A}\tilde{Q}^T(\tilde{D} + \tau_k I)^{-1}\tilde{Q}\tilde{A}^T \end{bmatrix}. \qquad (19)$$

Then, $x^{k+1}$ could be updated by

$$x^{k+1} = \tilde{Q}^T(\tilde{D} + \tau_k I)^{-1}\tilde{Q}(\tau_k v^k - \tilde{q}) - \tilde{A}^T v \qquad (20)$$

where

$$(\tilde{D} + \tau_k I)^{-1} = \begin{bmatrix} (d_1 + \tau_k)^{-1} & \cdots & 0 & \\ \vdots & \ddots & \vdots & \\ 0 & \cdots & (d_n + \tau_k)^{-1} & \\ & & & \tau_k^{-1} I_{m \times m} \end{bmatrix} \quad (21)$$

### 3.3. Adaptive Step Size

The ADMM algorithm is the first-order method with a linear convergence rate. For large-scale problems, achieving high accuracy requires more iterations. The adaptive step has shown a successful application in ADMM. Let $k$ be the current iteration and $k_0$ be an older iteration, such that $k_0 < k$. Let $\Delta u^k = u^k - u^{k_0}$, $\Delta H^k = x^k - x^{k_0}$. For $\hat{\alpha}_k, \hat{\beta}_k \in \mathbf{R}$, the optimal stepsize choice is then written as

$$\tau_{k+1} = (\hat{\alpha}_k \hat{\beta}_k)^{1/2}, \quad \hat{\gamma}_{k+1} = 1 + \frac{2\sqrt{\hat{\alpha}_k \hat{\beta}_k}}{\hat{\alpha}_k + \hat{\beta}_k}. \quad (22)$$

The spectral stepsize $\hat{\alpha}_k$ is updated by

$$\hat{\alpha}_k = \begin{cases} \hat{\alpha}_k^{MG} & \text{if } 2\hat{\alpha}_k^{MG} > \hat{\alpha}_k^{SD} \\ \hat{\alpha}_k^{SD} - \hat{\alpha}_k^{MG}/2 & \text{otherwise.} \end{cases} \quad (23)$$

where

$$\hat{\alpha}_k^{SD} = \frac{\langle \Delta u^k \Delta u^k \rangle}{\langle \Delta u^k, \Delta H^k \rangle}, \quad \hat{\alpha}_k^{MG} = \frac{\langle \Delta u^k, \Delta H^k \rangle}{\langle \Delta H^k, \Delta H^k \rangle} \quad (24)$$

The spectral stepsize $\hat{\beta}_k$ is similarly estimated by $\Delta G^k = z^k - z^{k_0}$ and $\Delta u^k = u^k - u^{k_0}$ [25]. To guarantee convergence, $\tau_k, \gamma_k$ are bounded by

$$\tau_{k+1} = \min\{\tau_{k+1}, (1 + C_{cg}/k^2)\tau_k\}, \quad (25)$$

$$\gamma_{k+1} = \min\{\gamma_{k+1}, 1 + C_{cg}/k^2\}, \quad (26)$$

where $C_{cg}$ is a (large) constant. Algorithm 1 summarizes all the steps.

---

**Algorithm 1** ADMM Solver

---

**Input:** initial values $v^0, z^0$

1: set $k = 0, k_0 = 0$
2: **repeat**
3:      $x^{k+1} \leftarrow$ solve the linear equation $\begin{bmatrix} \tilde{P} + \tau_k I & \tilde{A}^T \\ \tilde{A} & 0 \end{bmatrix} \begin{bmatrix} x^{k+1} \\ \nu \end{bmatrix} = \begin{bmatrix} \tau_k v^k - \tilde{q} \\ \tilde{b} \end{bmatrix}$
4:      **parallel for** $i = 1, \ldots, m + n$ **do**
5:      $x_i^{k+1} \leftarrow \gamma_k x_i^{k+1} - (1 - \gamma_k) z_i^k$
6:      $z_i^{k+1} \leftarrow \arg\min_{\tilde{z}_i} f_i(\tilde{z}_i) + \frac{\tau_k}{2}||x_i^{k+1} - v_i^k||^2$
7:      bound $z_i^{k+1}$ by (28)
8:      $\lambda_i^{k+1} \leftarrow \lambda_i^k + \tau_k(x_i^{k+1} - z_i^{k+1})$
9:      **end**
10:      **if** $\text{mod}(k, T_f) == 1$ **then**
11:         $(\tau_k, \gamma_k) \leftarrow \text{EstimateStep}(x^k, x^{k_0}, z^k, z^{k_0}, \lambda^k, \lambda^{k_0})$
12:         $k_0 \leftarrow k$
13:      **end if**
14:      $k \leftarrow k + 1$
15: **until** termination criterion is satisfied
16: **return** result

---

The function EstimateStep is stated in Algorithm 2. The estimation updates the stepsize $\tau_k$ and $\gamma_k$ with the frequency $T_f$. The proposed method works by assuming a local linear model for the dual optimization problem and selects an optimal stepsize. To guarantee convergence, a safeguarding method is adopted to ensure that excessive steps are not chosen if these linearity assumptions fail to hold.

---

**Algorithm 2** EstimateStep $(x^k, x^{k_0}, z^k, z^{k_0}, \lambda^k, \lambda^{k_0})$

1:  **parallel for** $i = 1, \ldots, m + n$ **do**
2:  $\Delta \lambda_i^k \leftarrow \lambda_i^k - \lambda_i^{k_0}$
3:  $\Delta H_i^k \leftarrow x_i^k - x_i^{k_0}$
4:  $\Delta G_i^k = z_i^k - z_i^{k_0}$
5:  **end**
6:  Compute spectral stepsizes $\hat{\alpha}_k, \hat{\beta}_k$ using (23)
7:  Update $\tau_{k+1}, \gamma_{k+1}$ using (22)
8:  Bound $\tau_{k+1}, \gamma_{k+1}$ using (25)
9:  **return** $\tau_{k+1}, \gamma_{k+1}$

---

*3.4. Termination Criteria*

For the given iterates $(x^k, z^k, u^k)$, the primal and dual residuals are defined as

$$r_{\text{prim}}^k = x^k - z^k$$
$$r_{\text{dual}}^k = \tau_k(v^k - v^{k-1})$$

It has been observed that these residuals approach zero as the algorithm approaches a true solution. The authors in [26] show that the pair $(u^k, z^k)$ satisfies optimality conditions for all $k > 0$. If the problem is also solvable, the residuals $r_{\text{prim}}^k$ and $r_{\text{dual}}^k$ will converge to zero. A termination criterion for detecting optimality is thus implemented by checking that $r_{\text{prim}}^k$ and $r_{\text{dual}}^k$ are small enough, i.e., $r_{\text{prim}}^k < \varepsilon_{\text{prim}}, r_{\text{dual}}^k = \varepsilon_{\text{dual}}$.

## 4. Acceleration Approaches

Another contribution of this paper is the GPU implementation of the proposed algorithms based on CUDA. In this section, we show how we parallized the proposed algorithm on CUDA and expose step by step the strategies that we used to achieve an optimal performance in our CUDA implementation of the proposed algorithm.

*4.1. Parallelization*

In the $k$-th iteration for updating $z^{k+1}$, we solve the nonlinear optimization problems (13). The procedure for solving each variable $z_i^{k+1} (i = 1, \ldots, n + m)$ is independent. We parallelize the steps for updating $z^{k+1}$ by

$$z_i^{k+1} = \arg \min_{l_i \leq \tilde{z}_i \leq u_i} f_i(\tilde{z}_i) + \frac{\tau_k}{2}||x_i^{k+1} - v_i^k||^2, i = 1, \ldots, n. \tag{27}$$

$$z_i^{k+1} = \arg \min_{\tilde{z}_i \geq 0} f_i(\tilde{z}_i) + \frac{\tau_k}{2}||x_i^{k+1} - v_i^k||^2, i = n+1, \ldots, n+m.$$

To avoid solving the constrained problems, we solve the unconstrained problems since the objective function is convex. $\tilde{z}_i^{k+1} = \arg \min_{\tilde{z}_i} f_i(\tilde{z}_i) + \frac{\tau_k}{2}||x_i^{k+1} - v_i^k||^2$ And $z_i^{k+1}$ is updated by

$$z_i^{k+1} = \Pi_{[l_i, u_i]}(z_i^{k+1}), i = 1, \ldots, n$$
$$z_i^{k+1} = \max\{\tilde{z}_i^{k+1}, 0\}, i = n+1, \ldots, n+m. \tag{28}$$

Listing 1 shows an overview of how we designed the kernel in the device on GPU. Each thread computes the new variables $z_i^{k+1}$ by (27) and (28) in parallel. To make the program more extendable, we created a base class named Cost to implement the different types of the cost function by the subclass in the device and pass the point to launch the kernel.

**Listing 1.** Computing the variables $z_i^{k+1}$ in parallel.

```
class Cost
{
public:
__device__ virtual double cost(double*z);
__device__ virtual double grad(double*z);
};
class FoF_cost: public Cost
{
public:
__device__ virtual double cost(double*z) override{
// compute the transaction cost
}
__device__ virtual double grad(double*z) override{
// compute the gradient
}
};
}
__global__ void update_z(double*z,{Parameter List}){
int tid = blockIdx.x * blockDim.x + threadIdx.x;
if (tid >= n + m)
return;
dfp_solver(&cost[tid], &z[tid]);
z[tid] = min(ub[tid],max(z[tid],lb[tid]));
return;
}
```

In our implementation, we apply the quasi-Newton method for minimizing (27), which is a second-order method that helps to quickly find the optimal value (see Listing 2).

**Listing 2.** Calling the quasi-Newton method (Davidon–Fletcher–Powell algorithm) in parallel.

```
__device__ void dfp_solver(Cost * cost, double *z){
double Hk = 1.0; double gk = 0.0;double dk = 0.0;
int k = 0;
while((++k) < MAXITER){
gk = cost->grad(z);
if((gk*gk) < epsilon)
break;
dk = -Hk * gk;
int m=0; double step = 1.0;
while( (++m) < INNERITER){
double cost = cost->cost(z + step * dk);
double cost_new = cost->cost(z) + step * gk * dk;
if( cost < cost_new){ break; }
else { step *= rho; };
}
double sk = step * dk;
*z += sk;
double yk = cost->grad(z) - gk;
Hk = sk / (yk);
}
}
```

**Remark 1.** *When the function $f_i(x), i = 1, \ldots, n$ is a linear or quadratic function, it is achievable to perform the matrix equilibration to transfer the matrix $\tilde{P}$ into diagonal matrix. After that, we*

*do not have any term like $\tilde{x}_i \tilde{x}_j$, which will reduce the number of iterations. However, for more complicated functions, such preconditioning will make updates of $z^{k+1}$ difficult to be parallelized.*

### 4.2. Do as Much as You Can on CUDA

CUDA is an extension of the C programming language created by NVIDIA. Its main idea is to have a large number of threads that solve a problem in parallel. To execute the program on GPU, we launch the kernels which are defined by the global keywords.

Before calling the kernel, CUDA needs the input data to be transferred from CPU to GPU through the PCI Express bus [27]. This stage of data transfer is a necessary part of any GPU code and can be the bottleneck affecting the program's performance. So once the data has been transferred to the device memory, it should not return to the CPU until all operations are finished. In our first CUDA implementation of the algorithm, this fact was not taken into account since the data is transferred from GPU to CPU during iterations. Therefore, the results of this first implementation are not outperform. This shows that direct implementations of not trivially parallelizable algorithms may initially disappoint the programmers expectations regarding GPU programming. This occurs regardless of the GPU used, which means that optimizations are necessary for this type of algorithms even when running on the latest CUDA architecture. We suggest finishing all the computing steps on GPU and avoid data copy as much as possible. First, we copy the matrix $\tilde{P}, \tilde{q}, \tilde{A}$ and $\tilde{b}$ to GPU and start the iteration. It is admirable to conduct the upgrade steps $(11)-(14)$ on GPU at each iteration; however, when it comes to the steps for judging the stopping criteria, we have to copy the data from GPU to CPU, which will increase unnecessary time on data transmission. Usually, the stopping criteria is calculated by comparing some matrix or vector norms, so we suggest avoiding data transfer by calling the cuBLAS to compute the norms and output the result on a CPU, which will significantly reduce the time spending on data transmission (see Listing 1).

### 4.3. Use CUDA Libraries

There exist excellent libraries shipped with the CUDA Toolkit that implement various functions on the GPU. We summarize the NVIDIA libraries used in our paper.

*cuBLAS* is a CUDA implementation of BLAS, which enables easy GPU acceleration of code that uses BLAS functions. We use the level-1 API function for the computation of norms. Level-2 and level-3 API functions are used for the matrix–vector product and matrix–matrix product (see Listing 3).

**Listing 3.** A general process for the proposed algorithm. The iteration is conducted on GPU and output of the residual is on CPU

```
double *A_host  // data in the host A is an n x n matrix
double *A, *x, *b, *r; // allocate the device memory
cudaMalloc((void *)&A, sizeof(double)*n*n);
cudaMemcpy(A, A_host, x_size, cudaMemcpyHostToDevice);
for (int i = 0; i < MAXITER; i++){
// excute the iteration
// compute the residual r = Ax-b
cudaMemcpy(r, b, n*sizeof(double), cudaMemcpyDeviceToDevice);
//the norm ||r|| is computed and storage on host
cublasDnrm2(handle, n, temp, 1, result);
if (*result < eps)
break;
}
```

*cuSolver* is a high-level package based on the cuBLAS and cuSPARSE libraries. It is a GPU accelerated library for decompositions and linear equation for both dense and sparse matrices. We use the *syevj* to compute the spectrum of a dense symmetric system.

### 4.4. Constant Memory and Page-Locked Memory Usage

Constant memory is a read-only memory, so it cannot be written from the kernels. Therefore, constant memory is ideal for storing data items that remain unchanged along with the whole algorithm execution and are accessed many times from the kernels. In our implementation of the proposed algorithm, we store the constant parameters we need during iteration. These values do not depend on the FOF model and do not change along with the thread execution, so they are ideal for constant memory. Our tests indicate that the algorithm is 5% to 15% faster, depending on the model size when using constant memory.

Page-locked (or "pinned") memory is used as a staging area to transfer the device to the host. We can avoid the cost of the transfer between pageable and pinned host arrays by directly allocating our host arrays in pinned memory. Pinned memory is beneficial because it avoids copying data directly between the CPU and GPU. Listing 4 shows how we apply fixed memory and compute residual norm in conjunction with the CuBLAS library. Since the residual is stored on the GPU, we need to calculate its norm to determine the stopping criteria on the CPU.

**Listing 4.** Using the CUDA memory and the CuBLAS library to compute residual norm.

```
void   Fof_solver ({ Parameter  List }) {
double  *resNorm;
// use the pinned memory
cudaMallocHost (( void **)&resNorm, sizeof (double));
for (int iter = 0; iter < MAXITER; ++iter) {
// excute the iteration
// compute the norm of the residual $r$
cublasDnrm2 (handle ,n,r ,1 ,resNorm );
if (*resNorm < eps ) {
break ;
}
}
```

**Listing 5.** Finding the optimal threads to achieve a maximum speedup.

```
#define THREAD 32
//...
dim3 block ((n−1)/THREAD + 1);
update_z<<<block ,THREAD>>>(z ,{ Parameter  List });
```

### 4.5. GPU Occupancy

Occupancy is the ratio of active warps per multiprocessor to the maximum number of possible active warps. The highest occupancy is no guarantee for obtaining the best overall performance, but the low occupancy always reduces the ability to hide latencies, resulting in general performance degradation. Therefore, we perform an experimental test on the device to determine exactly the best number of threads per block for our algorithm. The best values of serialized warps appear with a size of 128 threads and it achieves a maximum speedup. (See Listing 5)

Figure 1 shows the roofline of the kernel for computing the variables $z_i^{k+1}$. The roofline provides a visually intuitive way for users to identify performance bottlenecks and motivate code optimization strategies. Performance (GFLOP/s) is bound by

$$\text{GFLOP/s} \leq \min \begin{cases} \text{Peak GFLOP/s} \\ \text{Peak GB/s} \times \text{Arithmetic Intensity} \end{cases} \tag{29}$$

which produces the traditional Roofline formulation when plotted on a log-log plot. As can be seen from Figure 1, the performance of the kernel increases about $4.5\times$ after optimization when solving an FOF model ($N = 2000$). As the scale of the model increases, the utilization for compute and memory resource of the kernel approximates to the theoretical maximum performance bottleneck.

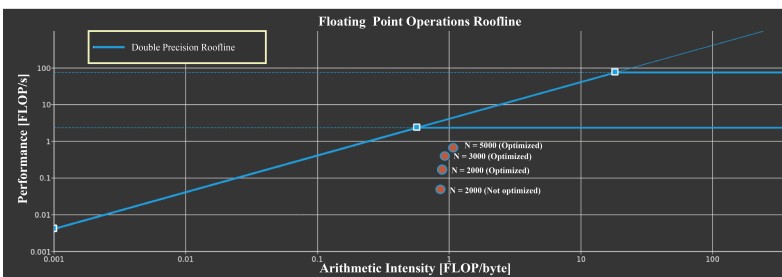

**Figure 1.** Overview of the utilization for compute and memory resources of the GPU presented as a roofline chart.

### 4.6. Storage Block Matrix on Device

In practice, we do not have to store the whole matrix $\tilde{P}, \tilde{A}, \tilde{b}$ and $\tilde{Q}$ on CUDA. It is feasible to store the matrix in blocks since those matrices have many zero blocks. All of the matrix–vector or matrix–matrix operations presented in our algorithm could be done via block operation. For example, $\tilde{Q}(\tilde{D} + \tau_k I)^{-1}\tilde{Q}(\tau_k v^k - \tilde{q})$ could be calculated by

$$\tilde{Q}^T(\tilde{D} + \tau_k I)^{-1}\tilde{Q} = \begin{bmatrix} Q^T & \\ & I \end{bmatrix}\begin{bmatrix} D_1 & \\ & \tau_k^{-1}I \end{bmatrix}\begin{bmatrix} Q & \\ & I \end{bmatrix}\begin{bmatrix} \tau_k v_1^k - q \\ \tau_k v_2^k \end{bmatrix} = \begin{bmatrix} Q^T D_1 Q \tau_k v_1^k - q \\ v_2^k \end{bmatrix}$$

where $v = [v_1, v_2], D_1 = \mathrm{diag}((d_1 + \tau_k)^{-1}, \dots, (d_n + \tau_k)^{-1})$. For the diagonal matrix, it is stored as an array in GPU. The computation steps of $D + \tau_k I$ could be done via vector–vector operation.

## 5. Experiment

We implemented our algorithms on CPU and parallelized them on GPU. All the programs run on a server with i9-10940X with 128 M RAM and one NVIDIA RTX2080ti, 11 GB memory. The server's CPU is i9-10940X, with 128 M memory and an NVIDIA RTX 2080ti GPU, 11 GB memory. The performance of the proposed methods is reported in comparison with the sequential quadratic programming (SQP) and the interior point method (IPM). We simulated the net value of $N = 5n$ different funds with $N$ ranging from 50 to 5000, which could be the maximum portfolio capacity for the funds. All the fund data is simulated by the Geometric Brownian motion (GBM), with different risk level $\sigma$ with regard to the different type of funds. The following table shows the running time for solving the problems (1) of SQP and the CPU implementation of our proposed method. The relative error is calculated by the $|x^* - x^*_{\mathrm{SQP}}|/x^*_{\mathrm{fmincon}}$, where $x^*$ and $x^*_{\mathrm{SQP}}$ represent the optimal value of the proposed method and the SQP, respectively. For smaller-scale problems, we ran numerical experiments 100 times and then averaged them, and for larger-scale problems, we ran the experiments 20 times.

**Table 1.** Running time and relative error for solving the portfolio problem.

| time(s)\$N$ | 10 | 50 | 100 | 150 | 200 |
|---|---|---|---|---|---|
| SQP | 0.018 | 0.106 | 0.236 | 0.614 | 0.939 |
| IPM | 0.038 | 0.145 | 0.355 | 0.780 | 1.088 |
| ADMM | 0.073 | 0.086 | 0.159 | 0.267 | 0.409 |
| relative error $1 \times 10^{-6}$ | 2.073 | 3.934 | 1.323 | 2.442 | 4.701 |
| $N$ | 300 | 350 | 400 | 450 | 500 |
| SQP | 1.798 | 2.267 | 2.735 | 3.814 | 5.203 |
| IPM | 2.376 | 3.144 | 3.894 | 5.000 | 7.024 |
| ADMM | 0.739 | 0.970 | 1.760 | 2.148 | 3.098 |
| relative error $1 \times 10^{-6}$ | 1.804 | 5.591 | 5.066 | 3.595 | 9.684 |
| $N$ | 700 | 800 | 900 | 1000 | 1200 |
| SQP | 12.297 | 19.300 | 26.657 | 51.833 | 118.145 |
| IPM | 13.992 | 17.669 | 25.677 | 36.335 | 82.592 |
| ADMM | 3.124 | 3.947 | 6.111 | 8.955 | 10.646 |
| relative error $1 \times 10^{-6}$ | 7.633 | 2.294 | 3.737 | 1.603 | 1.660 |
| $N$ | 1500 | 2000 | 3000 | 4000 | 5000 |
| SQP | 309.069 | 1082.843 | >3000 | >3000 | >3000 |
| IPM | 164.336 | 357.703 | 1683.123 | >3000 | >3000 |
| ADMM | 13.277 | 24.263 | 88.709 | 152.531 | 266.024 |
| relative error $1 \times 10^{-6}$ | 5.339 | 2.892 | 1.381 | | |

Table 1 shows the running time and relative error for solving the portfolio proble with different methods. As $N$ increases to more than $2 \times 10^3$, it will be challenge for the commercial solver to solve the problems in an acceptable time ($3 \times 10^3$ s). The proposed algorithm does not show an outstanding performance in the CPU version. Table 2 shows the GPU implementation of the proposed algorithm. By conducting the iteration in parallel, we reach a respectable acceleration.

**Table 2.** Running time for solving the portfolio problem in parallel.

| time(s)\$N$ | 10 | 50 | 100 | 150 | 200 |
|---|---|---|---|---|---|
| cuADMM | 7.075 | 0.096 | 0.125 | 0.151 | 0.201 |
| relative error $1 \times 10^{-6}$ | 2.568 | 3.367 | 1.324 | 2.403 | 1.775 |
| $N$ | 300 | 350 | 400 | 450 | 500 |
| cuADMM | 0.216 | 0.250 | 0.273 | 0.309 | 0.369 |
| relative error $1 \times 10^{-6}$ | 2.073 | 3.907 | 1.414 | 3.442 | 0.701 |
| $N$ | 700 | 800 | 900 | 1000 | 1200 |
| cuADMM | 0.613 | 0.623 | 0.700 | 0.703 | 0.871 |
| relative error $1 \times 10^{-6}$ | 1.041 | 3.042 | 1.400 | 2.705 | 0.799 |
| $N$ | 1500 | 2000 | 3000 | 4000 | 5000 |
| cuADMM | 1.255 | 1.539 | 2.958 | 4.751 | 5.733 |
| relative error $1 \times 10^{-6}$ | 1.042 | 3.041 | | | |

Figure 2 shows the calculation time and relative error curves of the FOF model (2.1) solved by different methods. We implement this method in CPU(ADMM) and GPU(cuADMM). We can see that the CPU implementation of the proposed method is better than SQP and IPM. When solving problems of different scales, the relative error of this method decreases faster, and the solving speed of the GPU version accelerates significantly.

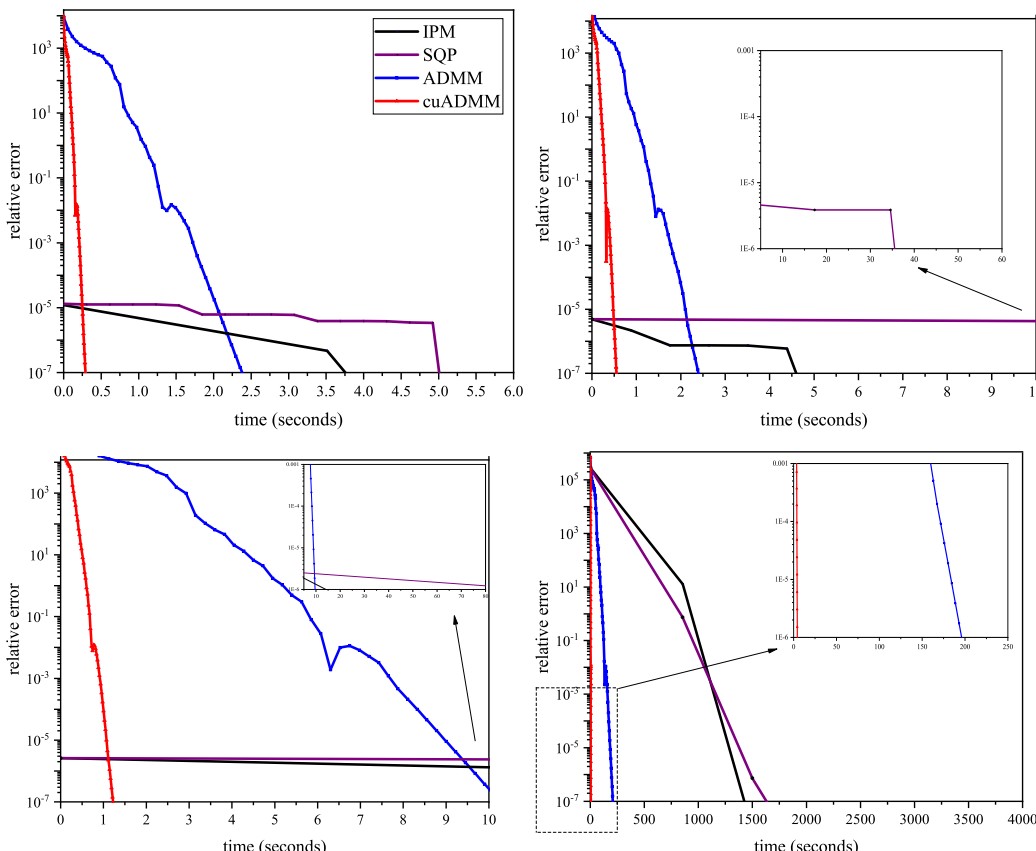

**Figure 2.** Comparison of different methods for $N = 500, 1000, 2000$ and $5000$. The parallelization of the proposed algorithm(cuADMM) outperforms and shows a significant acceleration compared with other methods.

Figure 3 (left) shows the changes of relative error for $N = 200, 500, 2000$ and $5000$. We also implemented a fixed stepsize ($\tau_k \equiv 1$) for comparison. For $N = 200$, the relative error of the fixed stepsize decreases very slowly . As the relative error decreases, the convergence speed becomes slower. The adaptive stepsize method converges and reaches the stopping criteria in only a few iteration.

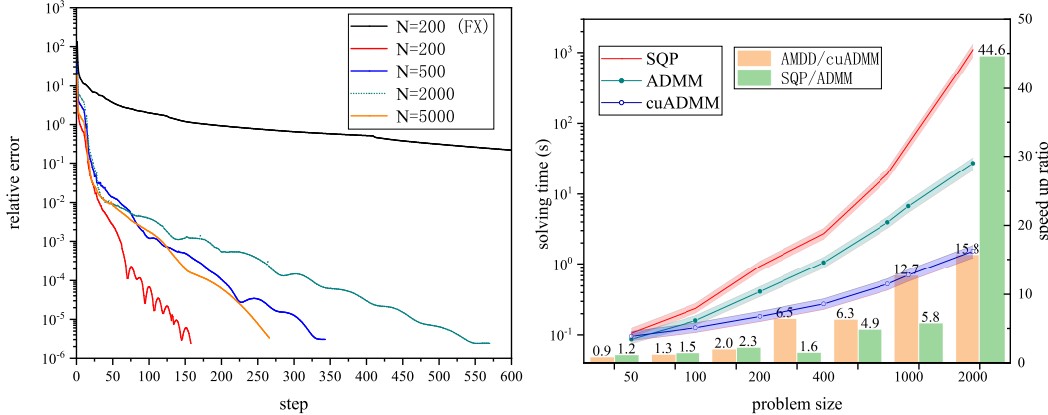

**Figure 3. Left:** The changes of relative error for different $N$. **Right:** Comparison of the speedup ratio for different problem sizes.

Figure 3 (right) compares the speedup ratios for problems of different sizes. When $N$ is small ($N < 200$), the acceleration of the GPU's implementation is not obvious as the copying data step takes up most of the time to execute the code. As $N$ increases,

the proposed algorithm not only maintains a fast convergence ratio but also significantly outperforms the existing methods.

## 6. Conclusions

This paper solved the convex FOF optimization model. The ADMM algorithm helped us separate the quadratic terms and the nonlinear terms of the objective function. We solved the KKT linear system and the nonlinear optimization problem at each iteration and parallelized the proposed algorithm on CUDA to solve nonlinear optimization problems. The number of iterations could be significantly reduced by the adaptive stepsize, which enables us to apply the proposed method to solve large-scale problems faster. We also implemented the proposed methods on CUDA and reported the optimization techniques to maximize the number of utilized kernels and use the device's memory architecture. The GPU version showed a very satisfying speedup for large-scale problems. Our numerical experiment raised the dimension of the FOF optimization model into a new scale, enabling the investors to allocate assets in the broader range of funds. However, there still exist certain limitations. For example, if there is a correlation between the nonlinear transaction cost functions when constructing the FOF model, the model could be more challenging to solve and parallelize. Therefore, the research on how to solve the model in parallel and reach the maximum performance of the algorithm on CUDA remains to be studied.

**Author Contributions:** Conceptualization, C.L. and Z.L.; methodology, Y.C. and C.L.; software, Y.C.; validation, C.L. and Y.C.; formal analysis, C.L. and Z.L.; investigation, C.L. and Y.C.; resources, Z.L.; data curation, Y.C.; writing—original draft preparation, Y.C. and C.L.; writing—review and editing, C.L. and Z.L.; visualization, Y.C.; supervision, Z.L.; project administration, C.L.; funding acquisition, Z.L. and C.L. All authors have read and agreed to the published version of the manuscript.

**Funding:** This research was funded in part by the National Natural Science Foundation of China grant number 61873254, in part by the China Postdoctoral Science Foundation grant number 2021M693226, and in part by the GHfund B grant number 20210702.

**Institutional Review Board Statement:** Not applicable.

**Informed Consent Statement:** Not applicable.

**Data Availability Statement:** Not applicable.

**Conflicts of Interest:** The authors declare no conflict of interest.

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
