# Peer review of "An ADMM Based Parallel Approach for Fund of Fund Construction"

_algorithms, doi:10.3390/a15020035_

Round 1

Reviewer 1 Report

The manuscript proposes a parallel algorithm for optimizing the fund of funds (FOF) problem, exploiting ADMM. The presentation keeps a balance between well-known and novel material. However, the authors should take care of the use of English and give a more descriptive title to their work. The title in the present form is general and doesn't reflect the actual contribution of the work.

Since a significant part of the contribution is the use of CUDA, a thorough implementation analysis should be given so that the readers may verify the authors' work. Also, the evaluation framework should be analyzed so that the readers may verify the proposed algorithm. The reference to a stochastic model for performing the evaluation is not adequate. 

In line 291, do the authors mean the Geometric Brownian motion (GBM), instead of the Geometry Brown Movement?

The authors should provide comparison results to other referenced works, especially regarding performance and relative error.  

The authors refer to the benefits of their algorithm; however, they should also report the limitations of their approach. The system of the evaluation phase is a high-end system. Considering that the NVIDIA Cuda architecture presents a homogeneous distribution of blocks and grids, the authors should report the optimization techniques to maximize the number of utilized kernels and use the device's memory architecture. The authors should clearly state the case they relied solely on the library they used (cublas) for those mentioned above. 

Author Response

Point 1: However, the authors should take care of the use of English and give a more descriptive title to their work. The title in the present form is general and doesn't reflect the actual contribution of the work.

Response 1: We gratefully appreciate for your valuable suggestion and we have proposed a revised title. After consideration, we renamed the title “An ADMM Based Parallel Approach for Fund of Fund Construction”, so as to better reflect our contribution.

Point 2: Since a significant part of the contribution is the use of CUDA, a thorough implementation analysis should be given so that the readers may verify the authors' work. Also, the evaluation framework should be analyzed so that the readers may verify the proposed algorithm. The reference to a stochastic model for performing the evaluation is not adequate.

Response 2: We gratefully appreciate your valuable suggestion. We have revised our paper, showed a detailed discussion of our CUDA implementation in Section 4.2 and provided the pseudo-code of our kernel function to show how we implemented our proposed algorithm and optimized the kernels on CUDA. We have presented our implementation analysis from line 208 to line 265. Moreover, we have highlighted all of the changes in our paper and we showed the evaluation framework in Section 4.1, Section 4.2, Section 4.3, and Section 4.4 from line 274 to line 360 so that it could be much easy for the reader to verify the proposed algorithm. In section 4.2, we show how we optimize the memory transfer between host and device. In section 4.3 and section 4.4 we show how we use the CUDA libraries, the constant memory and the page-locked memory to optimize the performance of the proposed algorithm. In section 4.5, a roofline of the kernel performance is shown in Fig.1. The source code has been provided in our attachment so that the readers may verify our work.

Point 3: In line 291, do the authors mean the Geometric Brownian motion (GBM), instead of the Geometry Brown Movement?

Response 3: Thank you for pointing this out. We mean the Geometric Brownian motion (GBM). We have corrected this error in line 377.

Point 4: The authors should provide comparison results to other referenced works, especially regarding performance and relative error.

Response 4: We gratefully appreciate for your valuable comment. We have provided the comparison results regarding to the performance(computing time) and relative error to the existing methods IPM and SQP. Since our FoF models are constructed according to the Chinese market rule and most researches use the commercial solvers to solve the FOF models. So we use the IPM and the SPQ methods as the state-of-art methods in comparision with the proposed algorithm. The comparision is provided and highlighted in Section 5, Fig.2.

Point 5: The authors refer to the benefits of their algorithm; however, they should also report the limitations of their approach.

Response 5: We gratefully appreciate for your valuable suggestion. We report the limitations of our approach in our conclusion:

Our numerical experiment raised the dimension of FOF optimization model into a new scale, enabling the investors to allocate assets in the broader range of funds. However, there still exist certain limitations. For example, if there is a correlation between the nonlinear transaction cost functions when constructing the FOF model, the model could be more challenging to solve and parallelize. Therefore, the research on how to solve the model in parallel and reach the maximum performance of the algorithm on CUDA remains to be studied.

Point 6: Considering that the NVIDIA Cuda architecture presents a homogeneous distribution of blocks and grids, the authors should report the optimization techniques to maximize the number of utilized kernels and use the device's memory architecture. The authors should clearly state the case they relied solely on the library they used (cublas) for those mentioned above.  

Response 6: We gratefully appreciate for your valuable comment. We have reported the optimization techniques from Section 4.2 to Section 4.5 and use the device's memory architecture in Section 4.4. The roofline (Fig.1) shows that as the scale of the model increases, the utilization for compute and memory resource of the kernel approximates to the theoretical maximum performance bottleneck.

Reviewer 2 Report

Authors should:

  • extend the literature overview and compare their findings with previous
  • extend their conclusion 
  • highlight limitations and recommendations for future research
  • extend practical implications of research results

Author Response

Response to Reviewer 2 Comments

Point 1: extend the literature overview and compare their findings with previous

Response 1: We gratefully appreciate for your valuable comment. We provide comparison results   regarding to the performance(computing time) and relative error to the existing methods IPM and SQP. Since our FoF models are constructed according to the Chinese market rule and most researches use the commercial solvers to solve the FOF models. So we use the IPM and the SPQ methods as the state-of-art methods in comparision with the proposed algorithm. The comparision is provided and highlighted in Section 5, Fig.2.

Point 2: extend their conclusion

Response 2: We gratefully appreciate your valuable suggestion. We extend our conclusion as follows:

This paper solved the convex FOF optimization model. The ADMM algorithm helped us separate  the quadratic terms and the nonlinear terms of the objective function. We solved the KKT linear system and the nonlinear optimization problem at each iteration and parallelized the proposed algorithm on CUDA to  solve nonlinear optimization problems. The number of iterations could be significantly reduced by the adaptive stepsize, which enables us to apply the proposed method to

solve large-scale problems faster. We also implemented the proposed methods on CUDA and  reported the optimization techniques to maximize the number of utilized kernels and use the device's memory architecture; The GPU version showed a very satisfying speed up for large-scale problems.  Our numerical experiment raised the dimension of FOF optimization model into a new scale, enabling the investors to allocate assets in the broader range of funds.

Point 3: highlight limitations and recommendations for future research

Response 3: We gratefully appreciate for your valuable suggestion. We highlight the limitations and recommendations for future research of our approach in our conclusion:

Our numerical experiment raised the dimension of FOF optimization model into a new scale, enabling the investors to allocate assets in the broader range of funds. However, there still exist certain limitations. For example, if there is a correlation between the nonlinear transaction cost functions when constructing the FOF model, the model could be more challenging to solve and parallelize. Therefore, the research on how to solve the model in parallel and reach the maximum performance of the algorithm on CUDA remains to be studied.

Point 4: extend practical implications of research results

Response 4: We gratefully appreciate for your valuable suggestion. We intend to implemt a practical investment realization in our paper.

 The implementation of a practical FoF model should give the input of the convariance matrix  and the expect return . However, these two parameters of a FoF model are generally estimated by the Black Litterman model, which is beyond the scope of our paper. So we mainly focus on computing the FoF model by simulated data and the optimization techniques to maximize the number of utilized kernels and use the CUDA device's memory architecture and we did not test the FoF model on a real-world dataset in our paper. Your suggestion is really helpful for our feature work.

Reviewer 3 Report

The paper considers a fund of fund optimization model which is an extension of mean-variance optimization by adding a different convex cost for each type of fund. The resulting model is a structured model that is convex and the authors propose an ADMM method to exploit the structure. The authors consider a relaxed ADMM with adaptive step size calculation and exploit the inherent parallelism to distribute the computation over different GPUS.

The amount of novelty is very moderate. The FOF model needs much more justification as a FOF strategy, no references are given other than stating that most FOF models are just some sort of variant of MVO. This is not at all true. In practice, most FOFs do NOT combine all different fund types and optimize a single model. Please justify the use of he FOF model presented in this paper.

In addition, the ADMM methods proposed are rather straightforward and adaptive step sizes have been known to be possibly effective for some time. I think the paper needs to clarify better what the main contributions of the paper are. 

Author Response

Response to Reviewer 3 Comments

Point 1: The amount of novelty is very moderate. The FOF model needs much more justification as a FOF strategy, no references are given other than stating that most FOF models are just some sort of variant of MVO. This is not at all true. In practice, most FOFs do NOT combine all different fund types and optimize a single model. Please justify the use of he FOF model presented in this paper.

Response 1: We gratefully appreciate for your valuable comment. Since the construction of a standard FOF model includes: determining the weight of each asset, choosing the fund managers

filtering the funds that show poor performance and so on. But those steps are out of this paper’s scope. In our model, we limit the fund type to equity funds where each of the funds we want to invest has been filtered. And we have re-stated the formulating of the problem in Section 2, which is highlighted in our paper. We mainly focus on computing the nonlinear optimization model and accelerating the proposed algorithm on CUDA.

Point 2: In addition, the ADMM methods proposed are rather straightforward and adaptive step sizes have been known to be possibly effective for some time. I think the paper needs to clarify better what the main contributions of the paper are.

Response 2: We gratefully appreciate your suggestion. Since a significant part of the contribution is the use of CUDA so we have provided a thorough implementation analysis and the techniques to optimize the number of utilized kernels and use the CUDA's memory architecture. We have revised our paper, showed a detailed discussion of our CUDA implementation in Section 4.2 and provided the pseudo-code of our kernel function to show how we implemented our proposed algorithm and optimized the kernels on CUDA. We have presented our implementation analysis from line 208 to line 265. Moreover, we have highlighted all of the changes in our paper and we showed the evaluation framework in Section 4.1, Section 4.2, Section 4.3, and Section 4.4 from line 274 to line 360 so that it could be much easy for the reader to verify the proposed algorithm. In section 4.2, we show how we optimize the memory transfer between host and device. In section 4.3 and section 4.4 we show how we use the CUDA libraries, the constant memory and the page-locked memory to optimize the performance of the proposed algorithm. In section 4.5, a roofline of the kernel performance is shown in Fig.1. The source code has been provided in our attachment so that the readers may verify our work.

Round 2

Reviewer 1 Report

In line 354 the starting letter seems to be 0 (zero) and not O.

The authors have significantly modified the manuscript to address the review comments 

Reviewer 3 Report

I am ok with the responses. Just do another round of English grammar checking.